# Integrating High-Intensity Interval Training into a School Setting Improve Body Composition, Cardiorespiratory Fitness and Physical Activity in Children with Obesity: A Randomized Controlled Trial

**DOI:** 10.3390/jcm11185436

**Published:** 2022-09-16

**Authors:** Meng Cao, Yucheng Tang, Yu Zou

**Affiliations:** 1Department of Physical Education, College of Sport, Shenzhen University, Shenzhen 518061, China; 2Department of Sport and Exercise Science, College of Education, Zhejiang University, Hangzhou 310027, China

**Keywords:** high-intensity interval training, obesity, children, school-based intervention, cardiorespiratory fitness

## Abstract

The aim of this study was to examine the effects of school-based high-intensity interval training (HIIT) on cardiorespiratory fitness and visceral adipose tissue (VAT) in children with obesity. A total of 40 students (11.0 ± 0.6 years; 20 boys) were randomized into an intervention group (IG) and control group (CG). The IG group performed a 12-week HIIT intervention with three sessions per week. Each session included 18 min of training (three sets of eight bouts of 15 s run at 100% maximal aerobic speed (MAS) separated by eight bouts of 15 s recovery run at 50% MAS) in PE class; the CG group were instructed to continue their normal behaviors. All subjects had indices of body mass index (BMI), fat mass (FM), body fat percentage (%BF), fat free mass (FFM), VAT, and maximal oxygen uptake (VO_2max_) measured at baseline and post-intervention. The cooperation of students was high, and all 40 students were included in the final analysis. A significant group–time interaction was determined in body composition (*p* < 0.05), with a significant decrease in BM (−3.4 ± 1.4 kg, *p* = 0.001; η^2^ = 0.63), BMI (−1.7 ± 0.5, *p* = 0.001; η^2^ = 0.58), %BF (−3.3 ± 1.4, *p* = 0.001; η^2^ = 0.54), and FM (−3.2 ± 1.4 kg, *p* = 0.001; η^2^ = 0.69), and VAT (−22.4 ± 9.8 cm^2^; *p* = 0.001; η^2^ = 0.61) in the IG. Furthermore, VO_2max_ exhibited a significant increase in the IG (4.5 ± 1.6 mL/kg/min, *p* = 0.001; η^2^ = 0.84) and CG groups (1.7 ± 1.1 mL/kg/min, *p* = 0.001; η^2^ = 0.44). Integrating regular school-based HIIT sessions is a suitable method to improve body composition, cardiorespiratory fitness, and physical activity in students with obesity. Trial Registration: ChiCTR2100048737.

## 1. Introduction

In the past 30 years, the prevalence of childhood obesity has increased significantly in different countries and regions [1]. Childhood obesity can be considered a probable early marker for adult obesity [2] and may increases the risk of cardiovascular disease but it is also associated with a significant reduction in cardiorespiratory fitness (CRF) [3]. Children who are overweight/obese have lower CRF and higher mortality than their normal-weight counterparts, and pediatric obesity is foremost a public health concern [4]. Meanwhile, obesity-induced excessive accumulation of visceral adipose tissue (VAT) may increase the risk of cardiovascular disease [5]. Therefore, increasing CRF and reducing VAT may be more critical for the obese population.

Regular physical activity (PA) during childhood plays a key role in reducing the risk of obesity and other non-communicable chronic diseases in children in the future. Unfortunately, over 41% of children and adolescents that are aged 6~17 years worldwide do not meet the current PA guidelines, which recommend at least 60 min of moderate–vigorous physical activity per day [6]. Hence, there is need for innovative and effective interventions to manage childhood obesity, promote their PA, and improve their physical fitness.

In recent decades, there has been renewed scientific interest in the efficacy of high-intensity interval training (HIIT) as a time-efficient way of improving health and managing obesity [7]. Growing evidence supported that HIIT could improve CRF (e.g., VO_2max_) and body composition [8,9]. Children spend most of their time in school, including physical activity and sedentary behavior [10]. Thus, schools are recognized as potential adequate settings for different exercise or multi-component interventions [11]. Previous systematic reviews and meta-analyses suggest that HIIT is feasible and time-efficient for improving cardiorespiratory fitness and body composition in adolescent populations [12,13]. However, extensive evidence supports the benefits of HIIT conducted in laboratory or hospital settings (treadmill or dynamometer interventions) [14,15]. Although a recent meta-analysis showed that school-based HIIT effectively improves several health outcomes in children [16], few studies have examined the efficacy of HIIT integration into PE classes for health promotion in obese children, and the results are also controversial [17,18].

This study aimed to determine the effect of integrating HIIT into school PE classes on body composition and cardiorespiratory fitness in obese children. The primary hypothesis of this study was that integrating HIIT into school PE classes improves body composition in terms of body fat percentage and VAT in children with obesity compared to a control group performing regular PE class activity. The secondary hypothesis was that cardiorespiratory fitness in terms of VO_2max_ would also improve.

## 2. Materials and Methods

### 2.1. Study Design and Participants

This study was designed as a parallel randomized controlled trial with two separate treatment arms: the intervention group (IG) and control group (CG), the allocation ratio was 1:1, each group consisted of 20 children with obesity. The participants were recruited from nine classes in one school, the number of subjects in each class ranged from 2 to 7. Prior to the study, the participants and their parents or legal guardians were informed of the purpose of the study, and their common written consent was obtained. The physical condition of the subjects was assessed by the PAR-Q questionnaire and their guardians were asked to ensure that there were no health problems [19]. In order to avoid the confounding interference of the rapid growth period on the results and essential obedience and understanding abilities that can better complete the training protocol, we selected students in the high age group of primary school as the subjects (10–13 years old). The students were eligible to participate if they (1) were aged 10 to 13 years; (2) provided written informed parental or guardian consent; (3) presented a BMI that was greater than or equal to the 95th percentile for their gender and age [20]; and (4) had no physical limitations to exercise (e.g., cardiac abnormalities, hypertension, diabetes, orthopedic, or neuromuscular disorders). Finally, 40 children with obesity (11.0 ± 0.6 years, 20 boys) were recruited from different classes in an elementary school, of whom 20 (11.2 ± 0.7 years) were randomly assigned to the HIIT intervention group (IG) and 20 (10.9 ± 0.4 years) to a control group (CG). Before the intervention, all the subjects participated in PE class four times a week and extracurricular sports daily. The trial was registered on the Chinese clinical trial registry (ChiCTR2100048737). The study was conducted according to the guidelines of the Declaration of Helsinki and approved by the medical ethics committee of the Department of medicine of Shenzhen University (PN-2020-045).

### 2.2. Sample Size

The sample size was calculated using G*Power software (Version 3.1; Dusseldorf, Germany), and was based on a previous study regarding VO_2max_ in obese adolescents (the effect size was 3.90) [21]. With a two-sides, 0.05 significance level, and VO_2max_ as the primary variable, 14 subjects in each group would allow us to detect a significant difference between groups at a Type I error rate of 5% and a power of 80%. Considering 20% dropouts, 34 subjects are required in total.

### 2.3. Randomization and Blinding

The randomly allocated sequence was computer (SPSS 20.0 (SPSS Inc., Chicago, IL, USA))-generated and sealed in sequentially numbered opaque envelopes. C.M generated the random allocation sequence, T.Y.C enrolled the participants, and Z.Y assigned the participants to interventions. This study is stratified by three age levels (10 years, 11 years, and 12 years), two genders (boys and girls), and three levels of BMI (21.0–23.0 kg/m^2^, 23.0–24.0 kg/m^2^, >24.0 kg/m^2^), with a total of 18 strata (3 × 2 × 3). As the subjects are enrolled, we determined the stratum to which they belong and were then separate randomized to either IG or CG (after baseline testing, the subjects were assigned using the next envelope in the sequence). BIA and VO_2max_ technicians were blinded to group allocation.

### 2.4. Anthropometrics and Body Composition

Body composition was the primary outcome of this study. The body composition was measured two days before the formal and second days after the intervention. The measurement was performed in the school classroom from 8:00 to 9:00 a.m., fasting for more than 10 h before measurements. There were two trained teachers (a male and a female) that measured, and one staff recording. The standing height (in cm to the nearest 0.5 cm) was measured without shoes using a wall-mounted scale. Body mass (BM), body mass index (BMI), body fat percentage (%BF), body fat mass (FM), fat-free mass (FFM), and visceral adipose tissue area (VAT) were analyzed by bioelectrical impedance analysis. BIA can be a reliable tool for measuring body composition and VAT; the reliability has been widely verified [22]. The Inbody 770 Body Composition Monitor (Biospace Co., Seoul, Korea) was used to obtain foot-to-foot BIA measures per the manufacturer’s guidelines, with participants standing barefoot on the footplates. Before the participants stepped on the scale, all the participants entered their age, gender, and height (cm). Furthermore, in order to ensure the accuracy of measurement, each subject was measured three times, and the average was calculated.

### 2.5. Cardiorespiratory Fitness

Cardiorespiratory fitness (CRF) was the secondary outcome of this study. CRF and maximal aerobic speed (MAS) was assessed by multistage 20-m shuttle run test (20-mSRT) in the school track. The 20-mSRT has been validated as a predictor of maximal aerobic capacity in children [23]. The participants performed the test in an outdoor track between two lines that were separated by 20-m, while keeping pace with the audio signals that were emitted from an MP3 that was produced by the National Coaching Foundation. School-age children and adolescents are familiar with the test method and the initial speed was set at 8.5 km/h and increased by 0.5 km/h every minute. The students were instructed to complete as many shuttles as possible. The test was ended if the participants gave up or failed to run 20-m run within the allotted time on two consecutive attempts. The speed at the last stage was considered as the MAS (km/h). After 20-mSRT, we use the formula by Mahar et al. [24] which was verified in children to calculate the maximum oxygen uptake (VO_2max_) of students. VO_2max_ (mL/kg/min) = 41.76799 + (0.49261 × laps) − (0.00290 × laps^2^) − (0.61613 × BMI) + (0.34787 × gender × age), 1 if boy or 0 if girl, and age in years. This test was conducted from 4:00 to 5:00 p.m. the day before the formal intervention, the fourth week, the eighth week, and the day after the intervention. A minimum of 2 subjects and maximum of 8 subjects were tested at the same time. Three trained teachers supervised and rated the test, and one staff member documented the results.

### 2.6. Heart Rate

Heart rate was monitored using a heart rate monitor (Polar team Oh1, Polar, Kemele, Finland). The chest belt of the Polar Team System was positioned in accordance with manufacturer instructions. The participants seated on a chair while the signal from the chest was checked for interference and signal quality. For resting heart rate (HR_rest_), simultaneous recorded during 5 min of silent rest was performed and selected the lowest value in this period as HR_rest_. The HR response was also monitored continuously during the test with a heart rate monitor. The peak heart rate (HR_max_) of each participant at the end of the test. HR_max_ was considered as the highest value based on 5-s averaged HR values.

### 2.7. Exercise Interventions

Obese students in the IG performed three high-intensity interval training (HIIT) sessions per week for 12 weeks. Except for the exercise tests and anthropometric measures, subjects in the IG and CG kept regular PE and activity in school without additional exercise training. The subjects were asked to maintain their current diet throughout the duration of the study. The HIIT sessions were integrated into the PE classes and after the regular physical education, the subjects went to the designated tracks for HIIT sessions (Figure 1). There were three study staff members that coordinated and supervised all training sessions. Before the intervention, the corresponding running distance according to the MAS of each subject was calculated and different marks were made on the track. The training sessions consisted of interval running and active recovery, conducted on the outdoor 300 m track. The training included three sections: (1) Standardized warm-up which included 5 min of moderate-intensity continuous jogging (~40–60% HRmax) and 5 min of dynamic stretching exercises, followed by 5*20-m sprints. (2) Formal training. According to the results of previous studies, combined with the convenience of practical operation, it is easier for obese children to complete and have better exercise adherence. We selected the following HIIT protocol. For the HIIT session, subjects were placed in different lanes of the track according to their MAS, performed three sets of eight 15-s bouts of high-intensity running (100% MAS, about 80~90% HR_max_) separated by eight 15-s active recovery (50% MAS) bouts with rest, 3-min rest between two sets, with a total duration time of 18-min. (3) The cool-down. At the end of the training session, subjects cooled down for about 5 min, running at low intensity and performing static stretching.

As shown in Figure 1, the distance that is required for each run can be calculated after measuring the participant’s MAS. The running sequence of subjects (MAS was 9 km/h) is as follows: First bout: A to B (15-s work period), and B to C to B (15-s recovery period), the next bout: B to A (15-s work period), and A to D to A (15-s recovery period). For example, a subject who had an MAS 9.0 km/h (2.5 m/s) had to complete 37.5 m in 15-s (i.e., 100% of MAS, that is A to B), which was followed by an active recovery to run over 18.8 m in 15-s [i.e., at 50% of MAS, that is B to C to B, the distance between B to C was 9.4 m]. During the running, subjects follow the music rhythm that is set in advance (including the countdown of 15 s). When the specified distance is completed within 15 s, the exercise intensity is deemed to be reached, and the A, B, C, and D were the marks on the track, we used cones of different colors.

### 2.8. Statistical Analysis

Statistical analysis was performed with the SPSS 20.0 (SPSS Inc., Chicago, IL, USA). Descriptive data are presented as the mean values ± standard deviation (SD). A Kolmogorov–Smirnov test was used to demonstrate that the data had a normal distribution (*p* > 0.05). A two-way analysis of variance (ANOVA) with repeated measures was used to compare values between the groups (IG and CG) at the two points (pre and post). A two-sided paired *t*-test was used to compare the differences of the changes before and after intervention (Δ). Post hoc analyses with Bonferroni’s correction were also performed. The effect size was measured by partial eta squared (η^2^). In the study, we considered that η^2^ from 0.01 to 0.06 are small, between 0.06 to 0.14 are medium, and above 0.14 are large, respectively [25]. *p*-values < 0.05 were considered statistically significant.

## 3. Results

A CONSORT diagram (Figure 2) summarizes participant flow through each stage of the trial and outlines the reasons for participant drop out. Of the 127 participants who met the inclusion and exclusion criteria, 40 (32.8%) were randomized. The other 87 participants were not randomized because of not signing the informed consent or other reasons. During the 12-week intervention period, no injuries were reported, and all the participants completed the training program. Finally, 40 obese children fully completed the current study.

A summary of the baseline characteristics for obese children is presented in Table 1. There were no significant differences between the groups at baseline. Table 2 and Figure 3 show the body composition and VO_2max_ of all the participants during the study and differences between pre- and post-intervention. The differences of VO_2max_ between the groups in IG and CG are presented in Figure 4.

A significant group and time interaction with a large effect was determined in body composition, with a significant decrease in BM (−3.4 ± 1.4 kg, *p* = 0.001; η^2^ = 0.63), BMI (−1.7 ± 0.5, *p* = 0.001; η^2^ = 0.58), %BF (−3.3 ± 1.4, *p* = 0.001; η^2^ = 0.54), and FM (−3.2 ± 1.4 kg, *p* = 0.001; η^2^ = 0.69) in the IG but not in the CG. A significant group and time interaction with a large effect was determined in the area of VAT, with a significant decrease (−22.4 ± 9.8 cm^2^; *p* = 0.001; η^2^ = 0.61) in the IG but not in the CG (+4.9 ± 12.7 cm^2^; *p* = 0.061; η^2^ = 0.09). Furthermore, the VO_2max_ was improved (+4.5 ± 1.6 mL/kg/min; *p* = 0.001; η^2^ = 0.84) in the IG and in the CG (+1.7 ± 1.1 mL/kg/min; *p* = 0.001; η^2^ = 0.44), revealing significant interaction effects and a large effect size. The increase of VO_2max_ in the IG group was greater than that in the CG group (*p* < 0.05). The HR_max_ and HR_rest_ of the subjects in the IG group decreased significantly when compared to the baseline value (−3.0 ± 3.2 b.p.m, *p* = 0.001; η^2^ = 0.29 and −2.7± 2.6 b.p.m, *p* = 0.001; η^2^ = 0.45, respectively), but there were no significant changes in HR_max_ and HR_rest_ in the CG group.

## 4. Discussion

The present study examined if a regular school-based HIIT intervention induced beneficial effects on the body composition and cardiorespiratory fitness of obese children. We presented evidence that a 12-week school-based HIIT program reduced BMI, FM, %BF, VAT, and improved VO_2max_ in obese children.

Body composition, especially fat content, is an important indicator that affects the health of obese children. The observed effects on FM and %BF reduction are consistent with previous school-based research by Bogataj et al. [26], who reported positive effects after 8-weeks of school-based HIIT and nutrition intervention in overweight adolescent girls. After 12-weeks of combined HIIT and dietary intervention, Plavsic et al. [27] reported a significant decrease in BMI and %BF in adolescent girls with obesity. Unlike the present study, the above combined dietary intervention or adopted different HIIT protocols (such as resistance training and treadmill). In contrast, some studies reported no changes in body composition of obese children after HIIT intervention and stated the lack of effect on body composition was due to the short intervention duration (<12 weeks) [28,29]. In comparison, da Silva and colleagues consider that this result was due to the unchanged diet rather than the total duration of the intervention [27]. Therefore, the present study showed that body composition improvements could also be reached with high exercise intensity interval training, and long-term (≥12 weeks) intervention may improve the body composition of obese children more effectively.

Evidence indicates that excessive visceral adipose tissue (VAT) may be more harmful than peripheral fat accumulation, increasing the risk of cardiovascular diseases and affecting adolescents’ cognitive function and brain health [30]. Meta-analysis suggested that aerobic exercise or combined diet and exercise interventions can reduce VAT in overweight/obese children and adolescents [31]. However, few studies focused on the effect of HIIT on VAT of children and adolescents. Our research team reported that running HIIT based on MAS as an intensity standard can significantly reduce the VAT of obese children [17]. The present study supported these consistent results, indicating that school-based HIIT can also significantly reduce VAT in children with obesity. Inconsistent results by Dias et al. have failed to detect significant decreases in VAT after HIIT in children with obesity [14]. Possible mechanisms of HIIT-induced fat loss include the production of catecholamines, increased fat oxidation and fat release from visceral fat storage, and increased oxygen consumption after exercise leading to an increase in fat reduction. In addition, a recent study has shown that interleukin-6 that is released from skeletal muscle after exercise stimulates lipolysis, thereby reducing visceral adipose tissue [32]. Future studies should observe the effects of HIIT on VAT in obese children and explore the possible mechanism further.

Cardiorespiratory fitness (CRF) is a powerful predictor of cardiometabolic disease outcomes in children and adolescents [33]. The improvement of HIIT on CRF in children with obesity has been supported by previously lab-based HIIT intervention studies [15,28]. A recent meta-analysis indicated that school-based HIIT effectively improves several health outcomes [16], but the integration and long-term effectiveness of HIIT interventions in the school setting remain questionable. Consistent with the previous systematic review and meta-analysis [9,34], the present study confirmed that CRF (VO_2max_) in obese children significantly increased following a 12-week HIIT intervention (+4.5 mL/kg/min), further supporting previous relevant school-based studies [17,26]. However, Plavsic et al. performed 4 × 4 min design, with work intensity at 85–90% HR_max_ HIIT protocol and failed to significantly improve the VO_2max_ of obese adolescents aged 13–19 years [27]. This inconsistency may be due to the difference in protocol design and load intensity. The work intensity in our study was set to 100% MAS, shorter bouts (15 s), and a higher average HR (165 b.p.m). In our series, this triggered an improvement of cardiorespiratory fitness. HIIT is also able to increase stroke volume that is induced by increased cardiac contractility and increase skeletal muscle diffusive capacity, thus improving aerobic capacity [7]. The significant decrease in the HR_rest_ and HR_max_ of the subjects in the IG group in this study also confirmed the improvement of cardiopulmonary fitness that may be induced by increased stroke output [35].

The strength of our study is exercise intervention that is based in the school setting, further verifying the feasibility of this intervention model for childhood obesity. This convenient and effective HIIT session model can be implemented in more schools. The limitations of this study included a small sample size which may affect the reliability of the results. The absence of a strictly controlled diet may cause confounding interference to the results, and the short intervention period cannot investigate the long-term effect on HIIT sessions. Another limitation was that younger children (<10 years) likely would not be suitable for training as suggested. Given that school-based HIIT was effective for weight management of obese children, and required short time, space, resources, organization, and supervision, the long-term effects of the program on physical activity levels and health benefits could be explored in future studies.

## 5. Conclusions

In conclusion, our findings indicated that integrating regular school-based HIIT sessions improved body composition and cardiorespiratory fitness in school children with obesity. Furthermore, the findings supported the implementation of HIIT as innovative PA programs in school settings, helping obese children develop a healthy and active lifestyle. More studies employing similar rigorous designs are needed to explore what modifications might be made to school-based HIIT programs to assist in weight management among children with obesity.

## Figures and Tables

**Figure 1 jcm-11-05436-f001:**
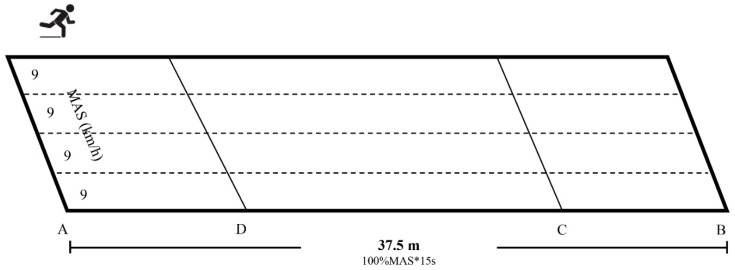
Schematic chart of running path. * means multiply.

**Figure 2 jcm-11-05436-f002:**
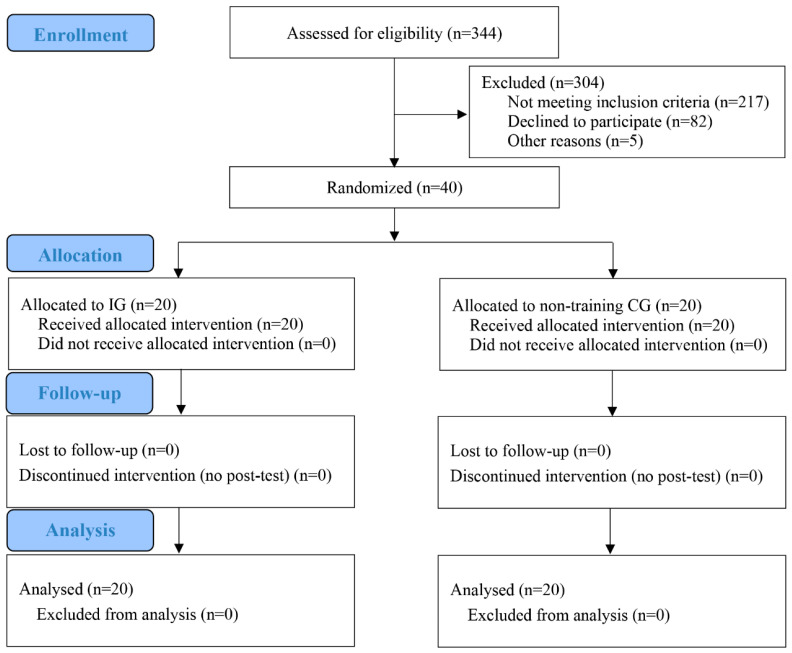
Consolidated standards of trials flow diagram.

**Figure 3 jcm-11-05436-f003:**
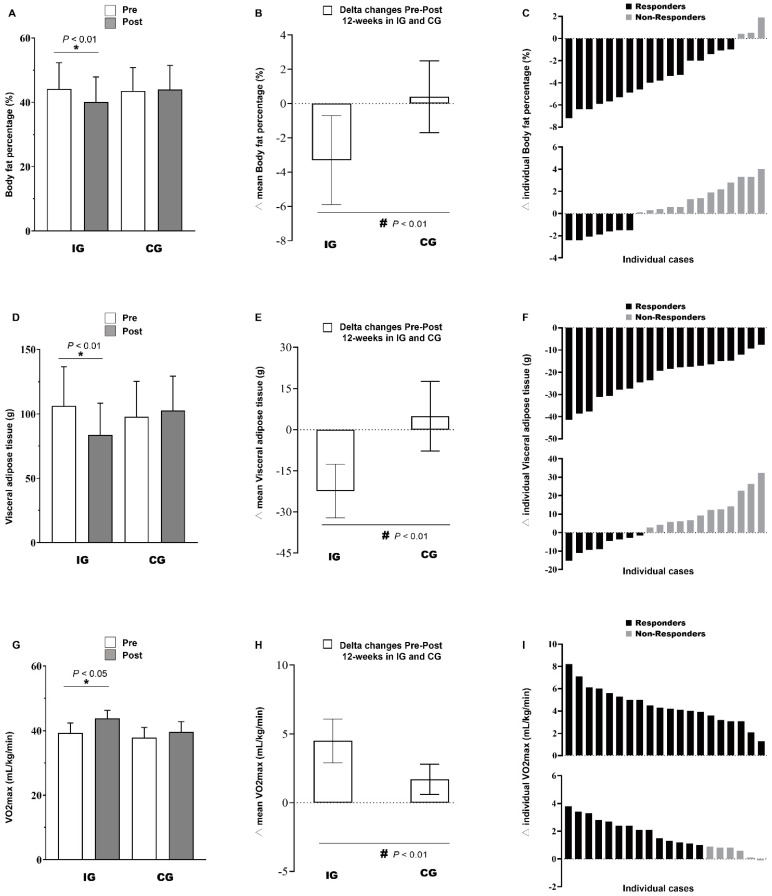
Pre-post changes (**A**,**D**,**G**), delta (mean) (**B**,**E**,**H**), and delta (individual) (**C**,**F**,**I**) of body fat percentage, visceral adipose tissue, and maximal oxygen uptake in children with obesity. * Denotes significant differences pre vs. post within group at level *p* < 0.05; # Denotes significant differences between IG vs. CG at level *p* < 0.05.

**Figure 4 jcm-11-05436-f004:**
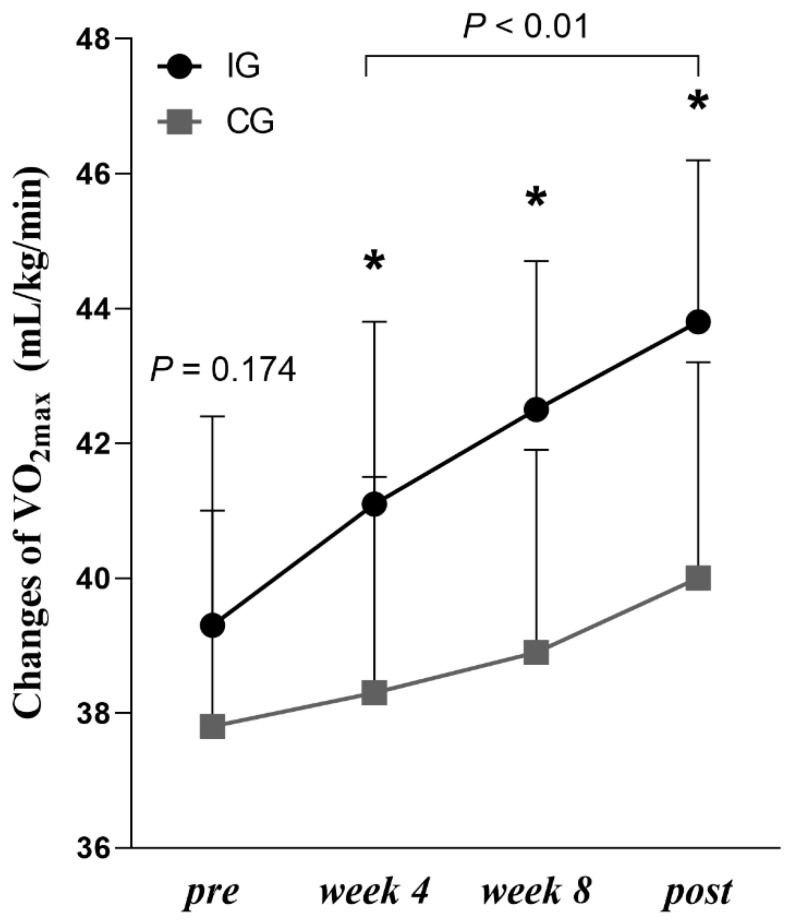
Differences of VO_2max_ between-group in IG and CG. * Denotes significant differences between IG vs. CG at level *p* < 0.05.

**Table 1 jcm-11-05436-t001:** Subject’s characteristics at baseline (Mean ± SD).

	Total (n = 40)	IG (n = 20)	CG (n = 20)
Boys/Girls (n)	20/20	10/10	10/10
Age (years)	11.0 ± 0.6	11.2 ± 0.7	10.9 ± 0.4
Height (m)	1.52 ± 0.07	1.53 ± 0.08	1.51 ± 0.05
Body Mass (kg)	54.6 ± 5.7	55.9 ± 6.9	53.3 ± 4.0
Body Mass Index (kg/m^2^)	23.6 ± 1.5	23.4 ± 1.6	23.8 ± 1.5

**Table 2 jcm-11-05436-t002:** Changes in the outcomes from pre- and post-intervention for the intervention group (IG) and control group (CG).

Outcomes	IG (n = 20)			CG (n = 20)			*p*-Value			Interaction η^2^
	Pre	Post	Δ	Pre	Post	Δ	Group	Time	Interaction	
BM (kg)	55.9 ± 6.9	52.5 ± 6.2 *	−3.4 ± 1.4 ^#^	53.3 ± 4.0	53.6 ± 4.3	0.2 ± 1.5	0.655	0.001	0.001	0.625
BMI (kg/m^2^)	23.4 ± 1.6	21.7 ± 1.5 *^, #^	−1.7 ± 0.5 ^#^	23.8 ± 1.5	23.5 ± 1.7	−0.3 ± 0.7	0.037	0.001	0.001	0.575
%BF (%)	44.2 ± 8.1	40.1 ± 7.8 *^, #^	−3.3 ± 2.6 ^#^	43.6 ± 7.2	44.0 ± 7.5	0.4 ± 2.1	0.497	0.001	0.001	0.541
FM (kg)	24.8 ± 6.0	21.2 ± 5.1 *	−3.2 ± 1.4 ^#^	23.3 ± 4.6	23.7 ± 5.0	0.4 ± 1.3	0.756	0.001	0.001	0.691
FFM (kg)	21.3 ± 5.0	21.3 ± 5.3	0.1 ± 1.5	21.2 ± 2.9	21.8 ± 3.1	0.6 ± 0.6	0.858	0.074	0.142	0.056
VAT (cm^2^)	106.2 ± 30.6	83.8 ± 24.6 *^, #^	−22.4 ± 9.8 ^#^	97.8 ± 27.5	102.7 ± 26.6	4.9 ± 12.7	0.539	0.001	0.001	0.605
20-mSRT (m)	483.0 ± 139.8	690.0 ±100.2 *^, #^	207.0 ± 90.4 ^#^	414.0 ± 126.0	500.0 ± 137.2 *	86.0 ± 60.2	0.002	0.001	0.001	0.395
VO_2max_ (mL/kg/min)	39.3 ± 3.1	43.8 ± 2.4 *^, #^	4.5 ± 1.6 ^#^	37.8 ± 3.2	39.6 ± 3.2 *	1.7 ± 1.1	0.004	0.001	0.001	0.505
HR_max_ (b.p.m)	201.4 ± 5.2	198.4 ± 5.3 *	−3.0 ± 3.2	200.3 ± 3.5	199.2 ± 4.4	−1.1 ± 3.7	0.849	0.053	0.089	0.074
HR_rest_ (b.p.m)	76.9 ± 4.0	74.3 ± 3.3 *	−2.7 ± 2.6 ^#^	75.6 ± 3.0	74.9 ± 2.1	−0.7 ± 1.6	0.624	0.023	0.005	0.186

Abbreviations: Pre, pre-intervention; Post, post-intervention; Δ, difference between pre- and post-intervention; 20-mSRT, 20 m shuttle run test; BM, body mass; BMI, body mass index; b.p.m beats per minute; %BF, body fat percentage; FM, fat mass, FFM, fat-free mass; HR heart rate; VAT, visceral adipose tissue; VO_2max_, maximal oxygen uptake. Significantly different within group post- vs. pre-intervention: * *p* < 0.05. Significantly different from control group: ^#^
*p* < 0.05.

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
