# Peer review of "Integrating High-Intensity Interval Training into a School Setting Improve Body Composition, Cardiorespiratory Fitness and Physical Activity in Children with Obesity: A Randomized Controlled Trial"

_jcm, 2022, doi:10.3390/jcm11185436_

Round 1

Reviewer 1 Report

In their current manuscript "Integrating High-intensity Interval Training into a School Setting Improve Body Composition, Cardiorespiratory Fitness and Physical Activity in Children with Obesity: A Randomized  Controlled Trial and 3-Month Follow-up Study”, Cao et al. applied a 12-weel school-based HIIT to investigate affects on body composition (i.e. fat mass/ visceral fat) and intermittent running performance in children.

The manuscript addresses an important topic (i.e. improving health-related fitness of children) and adds to the field. However, it lacks some novelty since a number of studies already investigated the topic (see 10.1371/journal.pone.0266427). I am not sure if the manuscript covers the main interest of the “Journal of Clinical Medicine”.

The work has several limitations and major revisions are needed before potential publication. This includes language correction as well as clarification of methodical issues.

Major points:

-        Clarify what your publication adds compared to other existing investigations (10.1371/journal.pone.0266427).

-        Trial registration: Pls. name the registered primary and secondary variables. Add a primary and secondary hypothesis accordingly.

-        The power calculation is useless without an effect size and a respective source for the ES.

-        Pls. provide a rational for the selected age range of 10 to 13 years.

-        Pls. provide the method for randomization. It appears that you also stratified the children by sex. If so, how was this done? How did you specifically select the 40 children you randomized? Any additional criteria?

-        The description of the actual intervention is a huge shortcoming of the study. It remains totally unclear what was actually done during training/ the intervention. First of all, how were the HIIT sessions controlled? How was made sure all sessions were fully performed? Second, how often did the children train (how many sessions per week)? What is meant by “The HIIT sessions were integrated into the PE classes, therefore, it cannot be carried out in three separate days”? How was intensity of the training controlled? It appears to be “all-out” sprints somehow connected to the 20-mST but how? What about the marks on the track? Would the children not simply run as fast as they can for 15 sec?

-        Pls. add a rational for why this sort of HIIT/SIT was selected.

-        You used a maximal exercise test for determination of CRF. Why was the 20-mST selected? For instance, the YYIR1C is a special test for children. However, in children in general, and those not used to exercise in particular, this can be problematic. How did you make sure they understood what “running to exhaustion” means (this is what you are supposed to do to have a valid test). Were children tested alone or in groups? How many raters were involved? Info on re-test is missing. Same setting, time of day etc? Was there any familiarization with the test performed? It might well be that the changes you observed only reflect a learning effect (indicated by the ctrl group). Pls. discuss.

-        I am not sure if the InBody is calibrated for children. Pls. provide this info and also provide data on repeated measures spec. for children (ICC, etc.)

-        Pls. indicate how “delta” was calculated (I believe on individual pre-post data). Two-sided paired t-test would be needed here. Pls. correct and recalculate if necessary.

-        Pls. add the HR data. This would largely improve the manuscript (also related to the above CRF discussion). Did heart rate change during the intervention?

-        It would be helpful to add a responder analysis for the main outcomes fat mass and exercise capacity. This could be a graphic showing how many children exceeded the typical error. Pls. also calculate the effect size.

-        The discussion appears to shallow. A more detailed comparison with other studies and specific findings is needed. Also, how do you suggest HIIT improved CRF and body composition. Why did it not work in other studies? The underlying mechanisms are important. To this respect, why is “…the required exercise intensity during these HIIT sessions in PE…” insufficient? Which adaptations are/are not triggered?

-        The strength and limitations need complete rewriting also in relation to the above.

Minor points:

-        I suggest to change the title. The intervention lasted 12 weeks, you do not actually have a 3-months FU performed after the intervention

-        Abstract: add the number of training sessions per week

-        Line 44 should read “Hence, there is need for innovative….”

-        Line 50 should read “Thus, schools are recognized…”

-        Line 66. Pls. change to “age 10 to 13 years”.

-        Pls. indicate how absence of limitations to exercise were assessed. Cardiac abnormalities appears very specific.

-        Line 71 should read “…randomly assigned to HIIT…”

-        I do not find any info on the control group except for the abstract. Pls. add.

-        Pls. add info on eating and additional exercise. Was there any instructions? Was is controlled? Pls. add to limitations if not.

-        Ref. #16 does not address validation of the 20-nSRT in children.

-        Line 105. Actually, if you sprint 8 times, there is only 7 recovery bouts. However, pls. add if a warmup or cool down phase was included. Sprinting without warmup appears a little odd.

-        The SD of height appears to low. How can it be at 0.05 to 0.08?

-        Pls. indicate the previous in-school and other activities of your participants.

-        Line 122 pls correct to “met”

-        Line 128, pls. change “are” to “is”

-        Line 136, pls. explain VAT on first mention. Again, is viszeral fat accurately measured in children by InBody?

-        Lines 139 to 142. Some redundancy here for the VO2max result.

-        Line 158.Should read “The observed effects on …. reduction…”

-        Line 161: “weeks”

-        Lines 162-163: Something is wrong with this sentence.

-        Line 167. “Believe” appears odd here.

-        Line 184: “Whether school-based HIIT interventions can improve CRF to a similar extent remains questionable…” Well, here is one meta-analysis: 10.1371/journal.pone.0266427

-        In the abstract you indicate that the “cooperation of students was high” and in the discussion “well-received by obese children”. How was this assessed? Were any scales used?

Author Response

Dear reviewer,

Thank you very much for your comments and professional advice. These opinions help to improve academic rigor of our manuscript. Based on your suggestion and request, we have made corrected modifications on the revised manuscript. We would like to show the details as follows:

Major points:

-        Clarify what your publication adds compared to other existing investigations (10.1371/journal.pone.0266427).

Response: We are so grateful for your kind question. First, compared with the existing investigations, this study only focuses on obese children, while the subjects of other studies are mostly regular children and adolescents. Second, the visceral adipose tissue area is the primary outcome indicator of this study, while other studies pay less attention to the impact of HIIT on this. Thirdly, it can be seen from this review (10.1371/journal.pone.0266427) that the existing HIIT research carried out based on school settings mainly adopts the 20-meter sprint as the primary form of training, and the present study adopts the maximum aerobic speed (MAS) determined according to each subject. We also revised the introduction section (page 2, lines 52-60).

-        Trial registration: Pls. name the registered primary and secondary variables. Add a primary and secondary hypothesis accordingly.

Response: Thanks for this kind recommendation. We registry our trial on the Chinese clinical trial registry, the trial registration ID was ChiCTR2100048737 (page 2, line 83). The primary and secondary variables were body composition (body fat percentage, fat mass, and visceral adipose tissue) and cardiorespiratory fitness (VO2max), respectively. The primary hypothesis is that integrating HIIT into a school setting favors body composition and cardiorespiratory fitness improvements in children with obesity. The second hypothesis is that integrating the HIIT session in school PE is possible and has long-term effects.

-        The power calculation is useless without an effect size and a respective source for the ES.

Response: Thanks, we agree with your suggestion but are uncertain about how to modify it. In the present study, the effect size was measured by partial eta squared (η2), and we considered that η2 from 0.01 to 0.06 are small, between 0.06 to 0.14 are medium, and above 0.14 are large, respectively.

-        Pls. provide a rational for the selected age range of 10 to 13 years.

Response: Thanks for your valuable feedback. Children aged 10-13 are in their early adolescence, which can exclude the confounding interference of rapid growth and development to a certain extent, making the investigation results more objective. Secondly, children of this age group have essential obedience and understanding abilities and can better complete the training protocol.

-        Pls. provide the method for randomization. It appears that you also stratified the children by sex. If so, how was this done? How did you specifically select the 40 children you randomized? Any additional criteria?

Response: Thanks, we appreciate this kind recommendation, and added the randomization methods on page 3, lines 96-102. The randomization process stratified by sex and initial BMI. We first screened children who met the BMI obesity standard from 18 classes in grades 4 to 6, then selected 20 boys and 20 girls who volunteered to participate in this study. In order to facilitate intervention, eligible subjects in the same class were selected first. Finally, the subjects participating in this experiment were distributed into nine classes, these are 403 (n=3), 404 (n=3), 501 (n=4), 502 (n=6), 504 (n=4), 505(n=6), 506 (n=7), 601 (n=5), and 604 (n=2). There are no other additional criteria.

-        The description of the actual intervention is a huge shortcoming of the study. It remains totally unclear what was done during training/ the intervention. First of all, how were the HIIT sessions controlled? How was made sure all sessions were fully performed? Second, how often did the children train (how many sessions per week)? What is meant by “The HIIT sessions were integrated into the PE classes, therefore, it cannot be carried out in three separate days”? How was intensity of the training controlled? It appears to be “all-out” sprints somehow connected to the 20-mST but how? What about the marks on the track? Would the children not simply run as fast as they can for 15 sec?

Response: Thank you for your questions. A detailed description of the training protocol is indeed essential. The children training three sessions per week. About this question “The HIIT sessions were integrated into the PE classes, therefore, it cannot be carried out in three separate days”, Usually, our exercise interventions should be arranged on three discontinuous days of the week. However, because the training intervention is integrated into PE class, it needs to intervene according to the time of PE class arranged by the school. The following figure and descriptions should answer the following questions.

 As shown in this figure, the distance required for each run can be calculated after measuring the participant's MAS. The running sequence of subjects (MAS was 8 km/h) as follow: First bout: A1 to B1 (15-s work period), and B1 to C1 to B1 (15-s recovery period), the next bout: B1 to A1 (15-s work period), and A1 to D1 to A1 (15-s recovery period). For example, a subject who had a MAS 9.0 km/h (2.5 m/s), he had to complete 37.5 m in 15-s (i.e., 100% of MAS, that is A2 to B2), which was followed by an active recovery to run over 18.8 m in 15-s [i.e., at 50% of MAS, that is B2 to C2 to B2, the distance between B2 to C2 (or A2 to D2) was 9.4 m]. During the running, subjects follow the music rhythm set in advance (including the countdown of 15 seconds). When the specified distance is completed within 15 seconds, the exercise intensity is deemed to be reached, and the A, B, C, and D were the marks on the track, we used cones of different colors.

We added this part according to your suggestion. Please see page 4-5, lines 144-176.

-        Pls. add a rational for why this sort of HIIT/SIT was selected.

Response: We are so grateful for your kind question. First, our previous findings consider running is a better intervention than cycling or other forms of HIIT/SIT. Second, to intervene in schools, considering the cost and operability, running is the most practical choice. Third, we consider whether the HIIT protocol can be applied to more schools in the future to help more obese teenagers manage their weight and improve their health.

-        You used a maximal exercise test for determination of CRF. Why was the 20-mST selected? For instance, the YYIR1C is a special test for children. However, in children in general, and those not used to exercise in particular, this can be problematic. How did you make sure they understood what “running to exhaustion” means (this is what you are supposed to do to have a valid test). Were children tested alone or in groups? How many raters were involved? Info on re-test is missing. Same setting, time of day etc? Was there any familiarization with the test performed? It might well be that the changes you observed only reflect a learning effect (indicated by the ctrl group). Pls. discuss.

Response: Thanks for your valuable feedback. We have added a description of the test details according to your comments (page 3, lines 130-133). First, the 20-mSRT is a widely used field test of aerobic fitness. It has been validated to predict maximal aerobic capacity in children and adolescents and is relatively safe (Léger et al., 1988; Mahar et al., 2011). Second, it is true that for some children, the 20-mSRT may not be suitable. However, in our region, the 20-mSRT is a regular test, and school-age children and adolescents are familiar with the test method. Combined with our previous research experience, we have not found any inappropriate situations. Of course, we still let every subject have a week of adaptive training before the test, including the adaptation to the rules and requirements of the 20-mSRT, to ensure that every subject is familiar with it. According to the test guidelines, the test ended if participants gave up or failed to run a 20-m run within the allotted time on two consecutive attempts. It also means that the subject has reached exhaustion. There are eight runways in the test field, so eight children can participate in the test together each time. A total of 4 raters participated in the test each time, responsible for recording the number of laps of children, monitoring whether they completed according to the rules, and encouraging them to complete as many as possible. The re-test is still conducted in the same setting, field, raters, and at the same time of the day.

-        I am not sure if the InBody is calibrated for children. Pls. provide this info and provide data on repeated measures spec. for children (ICC, etc.)

Response: Thank you for this valuable suggestion. This suggestion is very timely. We added details of measuring body composition using Inbody770 (page 3, lines 110-116). “The Inbody 770 Body Composition Monitor (Biospace Co., Korea) was used to obtain foot-to-foot BIA measures per the manufacturer's guidelines, with participants standing barefoot on the footplates. Before participants stepped on the scale, all participants entered their age, gender, and height (cm).”

-        Pls. indicate how “delta” was calculated (I believe on individual pre-post data). Two-sided paired t-test would be needed here. Pls. correct and recalculate if necessary.

Response: Thanks. Yes, we calculated the "delta" according to the difference of each individual pre- and post-intervention and then completed the statistical analysis using a Two-sided paired t-test. We also recalculated and checked the data.

-        Pls. add the HR data. This would largely improve the manuscript (also related to the above CRF discussion). Did heart rate change during the intervention?

Response: Your suggestion is very important, Thanks. We calculated and added the data on heart rate change and found that the resting heart rate of the subjects in the training intervention group decreased significantly. We added description of heart rate monitoring, the heart rate results, and discussed the possible reasons for this result.

Please see page 3-4, lines 134-143, page 6, lines 212-215, and page 8, lines 282-285.

-        It would be helpful to add a responder analysis for the main outcomes fat mass and exercise capacity. This could be a graphic showing how many children exceeded the typical error. Pls. also calculate the effect size.

Response: We appreciate for this kind recommendation, and we made a figure of the changes in the primary outcomes and added it to the manuscript. We calculated the effect size by partial eta squared (η2), as indicated in Table 2.

Figure 3. Changes in main outcomes

-        The discussion appears to shallow. A more detailed comparison with other studies and specific findings is needed. Also, how do you suggest HIIT improved CRF and body composition. Why did it not work in other studies? The underlying mechanisms are important. To this respect, why is “…the required exercise intensity during these HIIT sessions in PE…” insufficient? Which adaptations are/are not triggered?

Response: Thank you for this valuable suggestion. According to these suggestions, we revised the discussion section and added the potential mechanism of HIIT to improve CRF and body composition. Please see page 7, lines 209-212, page 8, lines 223-227, and lines 235-251.

-        The strength and limitations need complete rewriting also in relation to the above.

Response: Thanks for this kind recommendation. We revised the strength and limitations. Please see page 8, lines 284-290.

Minor points:

-        I suggest to change the title. The intervention lasted 12 weeks, you do not actually have a 3-months FU performed after the intervention

Response: Thanks. The original title did cause misunderstanding. We revised the title according to your suggestion. Please see the title on page 1, “Integrating High-intensity Interval Training into a School Setting Improve Body Composition and Cardiorespiratory Fitness in Children with Obesity: A Randomized Controlled Trial”.

-        Abstract: add the number of training sessions per week

Response: Thanks. We added “The IG group performed a 12-week HIIT intervention three sessions per week” in the abstract (page 1, line 13).

-        Line 44 should read “Hence, there is need for innovative….”

Response: Thanks. We revised this sentence (page 2, line 45).

-        Line 50 should read “Thus, schools are recognized…”

Response: Thanks. We revised this (page 2, line 51).

-        Line 66. Pls. change to “age 10 to 13 years”.

Response: Thanks. We changed “aged” to “age” (page 2, line 75).

-        Pls. indicate how absence of limitations to exercise were assessed. Cardiac abnormalities appears very specific.

Response: Thanks for your reminding. After the subjects signed the informed consent form, we assessed their physical condition by asking their guardians and filling in the PAR-Q questionnaire. (The PAR-Q is a simple self-screening tool that is typically used by fitness trainers or coaches to determine the safety or possible risks of exercising based on your health history, current symptoms, and risk factors). We added this description “The physical condition of subjects was assessed by the PAR-Q questionnaire and asked their guardians to ensure that there were no health problems.” (Page 2, lines 73-74).

-        Line 71 should read “…randomly assigned to HIIT…”

Response: Thanks. We’ve corrected “randomized” to “randomly” (page 2, line 80).

-        I do not find any info on the control group except for the abstract. Pls. add.

Response: Thanks for this kind recommendation. We added the info of CG “Except for the exercise tests and anthropometric measures, subjects in the CG kept regular PE and activity in school without additional exercise training” (page 4, lines 146-149).

-        Pls. add info on eating and additional exercise. Was there any instructions? Was is controlled? Pls. add to limitations if not.

Response: Thank you for this valuable suggestion. We added info on eating “Subjects were asked to maintain their current diet throughout the duration of the study.” (page 4, lines 146-149), and added to limitations.

-        Ref. #16 does not address validation of the 20-mSRT in children.

Response: Thanks. We’ve revised the right Ref (Mariana B, Edilson S, Miguel A. Validity of equations for estimating VO2peak from the 20-m shuttle run test in adolescents aged 11-13 years. J Strength Cond Res 2013, 27(10): 2774-81).

-        Line 105. Actually, if you sprint 8 times, there is only 7 recovery bouts. However, pls. add if a warmup or cool down phase was included. Sprinting without warmup appears a little odd.

Response: Thanks for this valuable suggestion. We added the training phases on page 4, lines 153-162.

-        The SD of height appears to low. How can it be at 0.05 to 0.08?

Response: Thank you for your questions. Because the height unit we use is cm, the SD is relatively low. We recalculated it, and the result is OK.

-        Pls. indicate the previous in-school and other activities of your participants.

Response: Thanks. We added this description of participants “Before the intervention, all subjects participated in PE class four times a week and extracurricular sports daily.” (page 2, line 81-83).

-        Line 122 pls correct to “met”

Response: Thanks. We’ve corrected “meet” to “met” (page 6, line 193).

-        Line 128, pls. change “are” to “is”

Response: Thanks. We’ve changed “are” to “is” (page 6, line 199).

-        Line 136, pls. explain VAT on first mention. Again, is viszeral fat accurately measured in children by InBody?

Response: Thanks. We mentioned VAT and the reliable of BIA on page 3, lines 110-112.

-        Lines 139 to 142. Some redundancy here for the VO2max result.

Response: Thank you for your comment. We revised it on page 6, lines 209-212.

-        Line 158.Should read “The observed effects on …. reduction…”

Response: Thanks. We revised this at page 7, line 232.

-        Line 161: “weeks”

Response: Thanks. We’ve changed “week” to “weeks” (page 7, line 235).

-        Lines 162-163: Something is wrong with this sentence.

Response: Thanks for your careful check. We revised this sentence on page 7, lines 236-238.

-        Line 167. “Believe” appears odd here.

Response: Thanks. We’ve changed “believe” to “consider” (page 8, line 241).

-        Line 184: “Whether school-based HIIT interventions can improve CRF to a similar extent remains questionable…” Well, here is one meta-analysis: 10.1371/journal.pone.0266427

Response: We gratefully appreciate for this valuable comment. We have revised the text to address your concerns and hope that it is now clearer. Please see page 8, lines 265-266.

-        In the abstract you indicate that the “cooperation of students was high” and in the discussion “well-received by obese children”. How was this assessed? Were any scales used?

Response: Thank you for your rigorous consideration. We explained this in the results section. During the 12-week intervention period, no injuries were reported, and all participants completed the training program. Finally, 40 obese children have fully completed the current study. Therefore, we consider that obese children can well accept HIIT sessions.

Reviewer 2 Report

Dear Editor,

Thank you for the opportunity to review the study conducted by Cao and colleagues. This is a randomized controlled study, designed to examine the effects of school-based high-intensity interval training on cardiorespiratory fitness and visceral adipose tissue in children with obesity. Despite the relevant theme, I have some questions that I present below.

·         At least 4 recent systematic reviews have already been published and should be considered in this section:

o   High-intensity interval training for improving health-related fitness in adolescents: a systematic review and meta-analysis

o   Effects of school-based high-intensity interval training on body composition, cardiorespiratory fitness and cardiometabolic markers in adolescent boys with obesity: a randomized controlled trial

o   Feasibility of incorporating high-intensity interval training into physical education programs to improve body composition and cardiorespiratory capacity of overweight and obese children: A systematic review

o   Effects of High-Intensity Interval Training and Moderate-Intensity Continuous Training on Cardiometabolic Risk Factors in Overweight and ObesityChildren and Adolescents: A Meta-Analysis of Randomized Controlled Trials

·         Considering the previous studies, the authors need to clarify what this study adds to the literature.

·         Please, specify the study hypotheses

·         Please, describe the study design as being parallel and include the allocation ratio.

·         The authors need to describe the setting and location where the data were collected

·         The parameters needed to calculate the sample size were not fully presented (e.g. the estimated difference between groups and the standard deviation). Please, describe the sample size in a separate subsection.

·         How many individuals per class were included in the study? How many classes? How many schools? This information was not included in the text.

·         Was the study protocol registered?

·         Please, describe the method used to generate the random allocation sequence

·         Also, describe the type of randomization and mechanism used to implement the random allocation sequence

·         Who generated the random allocation sequence, who enrolled participants, and who assigned participants to interventions

·         Were the participants, researchers and/or those who assessed the outcomes blinded?

·         What were the procedures adopted by the researchers before performing the bioelectric impedance? Please add a reference.

·         Please, specify primary and secondary outcome measures, including when they were assessed.

·         At what time of day was the intervention performed?

·         What about their routine outside the school? Were they advised to perform any other activity? What about their eating habits? This information needs to be included in the text.

·         Which variables were included in the LMMs? Did the authors consider random intercept and slope in the analysis using LMMs?

·         Please, describe the periods of recruitment and follow-up.

·         This is a randomized trial and a test for differences at baseline does not make sense. Please, remove the p-value column from table 1.

Author Response

Dear reviewer,

Thank you very much for your comments and professional advice. Those comments are all valuable and helpful for revising and improving our manuscript. Based on your suggestion and request, we have made corrected modifications to the revised manuscript. The primary corrections in the manuscript and the responses comments are as follows:

Despite the relevant theme, I have some questions that I present below.

  • At least 4 recent systematic reviews have already been published and should be considered in this section:

o   High-intensity interval training for improving health-related fitness in adolescents: a systematic review and meta-analysis

o   Effects of school-based high-intensity interval training on body composition, cardiorespiratory fitness and cardiometabolic markers in adolescent boys with obesity: a randomized controlled trial

o   Feasibility of incorporating high-intensity interval training into physical education programs to improve body composition and cardiorespiratory capacity of overweight and obese children: A systematic review

o   Effects of High-Intensity Interval Training and Moderate-Intensity Continuous Training on Cardiometabolic Risk Factors in Overweight and Obesity Children and Adolescents: A Meta-Analysis of Randomized Controlled Trials

  • Considering the previous studies, the authors need to clarify what this study adds to the literature.

Reply: Thanks, we appreciate this kind recommendation. We have carefully read the above articles and made corresponding supplements in the introduction section in combination with the actual situation of this study, and added the randomization methods on page 2, lines 52-60.

  • Please, specify the study hypotheses

Response: Thanks for your valuable suggestion. We added the study hypotheses “The primary hypothesis is that integrating HIIT into school PE classes favors body composition and cardiorespiratory fitness improvements in children with obesity” on page 2, lines 61-62.

  • Please, describe the study design as being parallel and include the allocation ratio.

Response: Thanks for your valuable feedback, and we added the description on page 2, lines 65-68.

  • The authors need to describe the setting and location where the data were collected.

Response: Thanks for this valuable suggestion. We described this in the research methods section (page 3, lines 105-108, and 130-133)

  • The parameters needed to calculate the sample size were not fully presented (e.g., the estimated difference between groups and the standard deviation). Please, describe the sample size in a separate subsection.

Reply: Thanks. We added the calculation of sample size in a separate subsection (page 2-3, lines 88-94). The VO2max was a primary variable in this study, so the sample size was calculated based on a previous study regarding VO2max in obese adolescents (Tjonna et al., 2009). With a two-sides, 0.05 significance level, and VO2max as the primary variable, 14 subjects in each group would allow us to detect a significant difference between groups at a type I error rate of 5% and a power of 80%. Considering 20% dropouts, 34 subjects are required to enter the intervention.

  • How many individuals per class were included in the study? How many classes? How many schools? This information was not included in the text.

Response: Thanks for this kind recommendation and we added on page 2, lines 67-68. Participants were recruited from one school. We first screened children who met the BMI obesity standard from 18 classes in grades 4 to 6, then selected 20 boys and 20 girls who volunteered to participate in this study. In order to facilitate intervention, eligible subjects in the same class were selected first. Finally, the subjects participating in this experiment were distributed into nine classes, these are 403 (n=3), 404 (n=3), 501 (n=4), 502 (n=6), 504 (n=4), 505(n=6), 506 (n=7), 601 (n=5), and 604 (n=2).

  • Was the study protocol registered?

Response: We are so grateful for your kind question. We registry our trial on the Chinese clinical trial registry, the trial registration ID was ChiCTR2100048737 (page 2, line 83).

  • Please, describe the method used to generate the random allocation sequence, Also, describe the type of randomization and mechanism used to implement the random allocation sequence, who generated the random allocation sequence, who enrolled participants, and who assigned participants to interventions. Were the participants, researchers and/or those who assessed the outcomes blinded?

Response: Thanks, we appreciate this kind recommendation, and added the randomization methods on page 3, lines 96-102. The randomly allocated sequence was computer (SPSS 20.0) generated and sealed in sequentially numbered opaque envelopes. C.M generated the random allocation sequence, T.Y.C enrolled participants, and Z.Y assigned participants to interventions. After baseline testing, participants were assigned to IG or CG using the next envelope in the sequence—the randomization process stratified by sex and initial BMI. BIA and VO2max technicians were blinded to group allocation.

  • What were the procedures adopted by the researchers before performing the bioelectric impedance? Please add a reference.

Response: Thank you for this valuable suggestion. This suggestion is very timely. We added details of measuring body composition using Inbody770 (page 3, lines 111-116), and we added a reference [22]. “The Inbody 770 Body Composition Monitor (Biospace Co., Korea) was used to obtain foot-to-foot BIA measures per the manufacturer's guidelines, with participants standing barefoot on the footplates. Before participants stepped on the scale, all participants entered their age, gender, and height (cm).”

  • Please, specify primary and secondary outcome measures, including when they were assessed.

Response: We gratefully appreciate for this valuable comment. We added the primary and secondary outcomes and the relevant details of these measurements (page 3, lines 104-108, and lines 130-133).

  • At what time of day was the intervention performed?

Response: Thank you for this kind questions. The HIIT sessions (3 times per week) were integrated into the PE classes, therefore, the specific intervention time will follow the time of each PE class. (The school has four PE classes a week.)

  • What about their routine outside the school? Were they advised to perform any other activity? What about their eating habits? This information needs to be included in the text.

Response: Thanks for this kind recommendation. We added this information on page 4, lines 146-149.

  • Which variables were included in the LMMs? Did the authors consider random intercept and slope in the analysis using LMMs?

Response: Thank you so much for your careful check. We did not clearly describe the statistical method and apologize for our error. We revised this on page 5, lines 182-183. The statistical method we used was a two-way analysis of variance (ANOVA) with repeated measures was used to compare values within each group (IG and CG) at the two points (pre and post), and we did not consider random intercept and slope.

  • Please, describe the periods of recruitment and follow-up.

Response: Thanks. We describe this on page 4, lines 145-146.

  • This is a randomized trial and a test for differences at baseline does not make sense. Please, remove the p-value column from table 1.

Response: Thank you for this valuable suggestion. We removed the p-value column from table 1.

Round 2

Reviewer 1 Report

This is a revised version and the authors have largely improved the manuscript. The intervention is now clearer and additional data has been provided. However, there is still room for improvement. What bothers me is that with the addition of text to the manuscript, the authors have included additional mistakes that now need revision. I recommend to avoid this by revising the manuscript internally before presenting it to reviewers.

Major:

- Hypothesis: Building the hypothesis is crucial. It should read “The primary hypothesis of this study was that integrating HIIT into school PE classes improves body composition in terms of [add outcome variable] in children with obesity compared to a control group performing regular PE class activity. The secondary hypothesis was that cardiorespiratory fitness in terms of [add outcome variable] would also improve.

- Power Calculation: pls. indicate the ES of the referenced study.

- Stratification. It is still unclear how stratification was performed. This is crucial and should be explained with necessary care.

- you have still not provided evidence that the respective device (Inbody 770 Body) is calibrated for children and what the values for reproducibility are.

- 20 MSRT. Pls. add the information that “school-age children and adolescents are familiar with the test method” and indicate that a familiarization session was not included. I also learned that the test was performed “the day before the formal intervention, the fourth week, the eighth week, and the day after the intervention”. This is important information. I strongly suggest to add this data in a separate figure documenting the change in test performance over time per group with between-group differences reported.

- It is appreciated that you added a figure and legend for explanation. I wonder why this was not done from the beginning. However, you have included two samples. Pls. stick with either the 8 or 9 kmh MAS example.

- Nice to have figure 3. However, it is missing an appropriate legend with abbreviations, indication of performed tests, indication of p values and the fact that individual data is shown. Pls. also differentiate for between-group and within-group comparison (the latter is missing).

- Again, pls. include a responder analysis. Currently, we do not know how many of the IG children improved above typical error. Use method described here “Alvarez C, Ramírez-Campillo R, Ramírez-Vélez R, Izquierdo M. Effects of 6-Weeks High-Intensity Interval Training in Schoolchildren with Insulin Resistance: Influence of Biological Maturation on Metabolic, Body Composition, Cardiovascular and Performance Non-responses. Front Physiol. 2017 Jun 29;8:444.”

- Selection of age group. Pls. include your explanation in the methods AND the limitation section as younger children likely will not be suitable to be training as suggested.

- The manuscript is still missing a rational for why this sort of HIIT/SIT was selected. Why this intensity, duration, and number of sessions per week? Is there a study showing for example, that 15 sec runs are better/different than 4x30 sec runs

Minor comments:

Abstract: pls change to “The IG group performed a 12-week HIIT intervention with three sessions per week.”

Line 59: should read “for health promotion in obese children”

Line 67: pls. change to “each group consisted of 20 children with obesity”

Line 67-68: It is unclear what “…in each class ranged from 2 to 7.” means.

Line 80: pls. change to “…were randomly assigned to the HIIT intervention group (IG)…”

Line 90: pls. change to “…and was based on a..”

Line 107: Why were two teachers measured?

Line 132: should read “A maximum of 8 subjects were test at the same time, three trained teachers supervised and rated the test, and one staff member documented the results.” Pls. also indicate the minimum number of participants.

Line 138: should read “For resting heart rate (HRrest), simultaneous recorded during 5 minutes of silent rest was performed”. Pls. also indicate if the lowest or the mean HR of that period was selected. What is meant by “HR data were later analyzed to compute the average HR response during training across the 12-week intervention.”? You only present static HR data.

Line 145: should read “Obese students in the IG performed three high-intensity interval training (HIIT) sessions per week for 12 weeks”.

Line 154: “included”

159: “high-intensity running”

Line 160: what are “15-s recovery bouts with rest”? Its rather “active recovery” at which intensity?

Line 183: “within each group” is wrong. This is between-group.

Line 209: “VO2max was improved… in the IG and in the CG…”

Table 2: HR and bpm are missing in the legend.

Line 269: it is not “12-week HIIT sessions” but “a 12-week HIIT intervention”

Line 273: pls. correct to “The work intensity in our study was set to 100% MAS, shorter bouts (15s), and a higher average HR (165 b.p.m). In our series, this triggered an improvement of cardiorespiratory fitness.” Next sentences need revision, too. What are the authors trying to say here? “The greater increase of VO2max caused by HIIT maybe because this training can induce adaptation of central (cardiovascular) and peripheral (skeletal muscle) [36]. Such as HIIT can increasing the mitochondrial density, thus producing more ATP for working muscles [37].

Line 284: Pls. remove redundant sentence “Embedded HIIT within the school PE class, so it was integrated into daily school life.” Also, this is not a new model.

Line 291: You cannot state it was well-received since you did not assess this.

Author Response

Dear reviewer,

Thank you again for your careful comments and professional advice. These comments help to improve the academic rigor of our manuscript. According to these suggestions, we corrected the manuscript again. We want to show the following details:

Major:

- Hypothesis: Building the hypothesis is crucial. It should read “The primary hypothesis of this study was that integrating HIIT into school PE classes improves body composition in terms of [add outcome variable] in children with obesity compared to a control group performing regular PE class activity. The secondary hypothesis was that cardiorespiratory fitness in terms of [add outcome variable] would also improve.

Response: Thanks for this kind recommendation. Following your suggestion, we revised the hypothesis description on page 2, lines 61-65.

- Power Calculation: pls. indicate the ES of the referenced study.

Response: Your suggestion is very important, Thanks. We calculated and added the ES of the referenced study (ES = 3.90) on page 3, lines 94-95.

- Stratification. It is still unclear how stratification was performed. This is crucial and should be explained with necessary care.

Response: Thank you for this valuable suggestion. We describe the stratification process as "This study stratifies by three age levels (10 years, 11 years, and 12 years), two genders (boys and girls), and three levels of BMI (21.0-23.0 kg/m2, 23.0-24.0 kg/m2, > 24.0 kg/m2), have a total of 18 strata (3×2×3). As subjects are enrolled, we determine the stratum to which they belong and then make from separate randomized to either IG or CG (After baseline testing, subjects were assigned using the next envelope in the sequence)" on page 3, lines 103-108.

- you have still not provided evidence that the respective device (Inbody 770 Body) is calibrated for children and what the values for reproducibility are.

Response: The device has not been specially calibrated for children, but its accuracy and effectiveness have been verified. Furthermore, in order to ensure the accuracy of measurement, each subject was measured three times, and the average was calculated (page 3, lines 123-124). The InBody 770 analyzers utilize a tetrapolar 8-point tactile 160 electrode system, and take 30 impedance measurements with six frequencies (1 kHz, 5 kHz, 50 kHz, 250 kHz, 500 kHz, and 1000 kHz) with a test duration of approximately 60 seconds (doi:10.1016/j.jocd.2018.10.008). The Inbody 770 has no empirical estimations based on age, sex, ethnicity, or body type. Instead, Direct Segmental Multi-Frequency BIA technology measures body segments separately for an accurate analysis based on the subject's unique body. InBody devices showed a “high correlation with the DEXA results” for assessing appendicular lean mass (β ≥ 0.95), fat-free mass (R2 ≥ 0.95), and percent body fat (R2 ≥ 0.89) (doi: 10.3390/life12070994).

- 20 MSRT. Pls. add the information that “school-age children and adolescents are familiar with the test method” and indicate that a familiarization session was not included. I also learned that the test was performed “the day before the formal intervention, the fourth week, the eighth week, and the day after the intervention”. This is important information. I strongly suggest to add this data in a separate figure documenting the change in test performance over time per group with between-group differences reported.

Response: We appreciate for this kind recommendation. We add this information on page 3, lines 131-132, and we made a figure of in test performance over time per group with between-group differences reported and added it to the manuscript.

Figure 4.

- It is appreciated that you added a figure and legend for explanation. I wonder why this was not done from the beginning. However, you have included two samples. Pls. stick with either the 8 or 9 kmh MAS example.

Response: Thank you so much for your careful check. We redraw Figure 1 regarding your suggestion and revised this description (page 5, lines 181-190).

- Nice to have figure 3. However, it is missing an appropriate legend with abbreviations, indication of performed tests, indication of p values and the fact that individual data is shown. Pls. also differentiate for between-group and within-group comparison (the latter is missing). Again, pls. include a responder analysis. Currently, we do not know how many of the IG children improved above typical error. Use method described here “Alvarez C, Ramírez-Campillo R, Ramírez-Vélez R, Izquierdo M. Effects of 6-Weeks High-Intensity Interval Training in Schoolchildren with Insulin Resistance: Influence of Biological Maturation on Metabolic, Body Composition, Cardiovascular and Performance Non-responses. Front Physiol. 2017 Jun 29;8:444.”

Response: We appreciate for this kind recommendation and reference article. We redraw Figure 3 regarding this article (page 8).

Figure 3. Pre-post changes (A,D,G), delta (mean) (B,E,H), and delta (individual) (C,F,I) of body fat percentage, visceral adipose tissue and maximal oxygen uptake in children with obesity. *Denotes significant pre-post intragroup changes at level P < 0.05. #Denotes significant differences between IG vs. CG at level P < 0.05.

- Selection of age group. Pls. include your explanation in the methods AND the limitation section as younger children likely will not be suitable to be training as suggested.

Response: Thanks for your valuable feedback. We explanation this in the methods “In order to avoid the confounding interference of the rapid growth period on the results and essential obedience and understanding abilities that can better complete the training protocol, we selected students in the high age group of primary school as the subjects (10-13 years old)” on page 2, lines 75-78. We added this in the limitation section on page 10, lines 308-310.

- The manuscript is still missing a rational for why this sort of HIIT/SIT was selected. Why this intensity, duration, and number of sessions per week? Is there a study showing for example, that 15 sec runs are better/different than 4x30 sec runs

Response: Thank you for your rigorous consideration. Our previous meta-analysis pooled 18 studies, showed that running HIIT three times a week for more than eight weeks improved children's cardiorespiratory fitness. Considering a practical application, compared with the %HRmax and %VO2max, it is easier to use the %MAS as the intensity standard basis in the school setting. It is easier to operate in the playground in a shorter time (15 s) than in a long time (30 s). Moreover, for obese children, shorter work time is easy to complete and can improve exercise adherence. We added this description on page 4, lines 168-171.

Minor comments:

Abstract: pls change to “The IG group performed a 12-week HIIT intervention with three sessions per week.”

Response: Thanks. We revised this at abstract (page 1, line 12).

Line 59: should read “for health promotion in obese children”

Response: Thanks. We revised “on” to “in” (page 2, line 58).

Line 67: pls. change to “each group consisted of 20 children with obesity”

Response: Thanks. We changed “have” to “consisted of” (page 2, line 70).

Line 67-68: It is unclear what “…in each class ranged from 2 to 7.” means.

Response: Thank you so much for your careful check. We revised it as “The number of subjects recruited in each class ranged from 2 to 7” (page 2, line 71).

Line 80: pls. change to “…were randomly assigned to the HIIT intervention group (IG)…”

Response: Thanks. We added “were” (page 2, line 84).

Line 90: pls. change to “…and was based on a..”

Response: Thanks. We added “was” (page 2, line 94).

Line 107: Why were two teachers measured?

Response: Thank you for your questions. Considering privacy, we arranged for a male teacher and a female teacher to measure boys and girls, respectively. For example, it is more convenient to wear a heart rate belt.

Line 132: should read “A maximum of 8 subjects were test at the same time, three trained teachers supervised and rated the test, and one staff member documented the results.” Pls. also indicate the minimum number of participants.

Response: Thanks for your careful check. We have revised the text as your comments (page 3-4, lines 141-143).

Line 138: should read “For resting heart rate (HRrest), simultaneous recorded during 5 minutes of silent rest was performed”. Pls. also indicate if the lowest or the mean HR of that period was selected. What is meant by “HR data were later analyzed to compute the average HR response during training across the 12-week intervention.”? You only present static HR data.

Response: Thanks for this kind recommendation. We revised this sentence (page 4, lines 149-150). This sentence “HR data were later analyzed to compute the average HR response during training across the 12-week intervention” causes ambiguity, so we deleted it.

Line 145: should read “Obese students in the IG performed three high-intensity interval training (HIIT) sessions per week for 12 weeks”.

Response: Thanks for your careful check. We added “the” and deleted “in” (page 4, line 156).

Line 154: “included”

Response: Thanks. We changed “includes” to “included” (page 4, line 165).

159: “high-intensity running”

Response: Thanks. We revised it (page 4, line 172).

Line 160: what are “15-s recovery bouts with rest”? Its rather “active recovery” at which intensity?

Response: Thank you for your questions. We added “active recovery (50% MAS)” on page 4, line 173.

Line 183: “within each group” is wrong. This is between-group.

Response: Thanks for your careful check. We changed “within each group” to “between-group” on page 5, line 196.

Line 209: “VO2max was improved… in the IG and in the CG…”

Response: Thank you for this kind recommendation. We revised as your suggestion on page 7, lines 232-233.

Table 2: HR and bpm are missing in the legend.

Response: Thanks. We added this in Table 2.

Line 269: it is not “12-week HIIT sessions” but “a 12-week HIIT intervention”

Response: Thanks. We revised this on page 10, line 287.

Line 273: pls. correct to “The work intensity in our study was set to 100% MAS, shorter bouts (15s), and a higher average HR (165 b.p.m). In our series, this triggered an improvement of cardiorespiratory fitness.” Next sentences need revision, too. What are the authors trying to say here? “The greater increase of VO2max caused by HIIT maybe because this training can induce adaptation of central (cardiovascular) and peripheral (skeletal muscle) [36]. Such as HIIT can increasing the mitochondrial density, thus producing more ATP for working muscles [37].

Response: Thanks for your valuable feedback. We revised this sentence on page 10, lines 291-293. The underlying mechanism described in the following sentence may not be relevant to this study, so we deleted it.

Line 284: Pls. remove redundant sentence “Embedded HIIT within the school PE class, so it was integrated into daily school life.” Also, this is not a new model.

Response: Thanks for this valuable suggestion. We removed this sentence and revised the description (page 10, lines 303-304).

Line 291: You cannot state it was well-received since you did not assess this.

Response: Thanks. We deleted “well-received” and revised this sentence on page 10, line 310-311.

Reviewer 2 Report

Dear authors,

Before the hypotheses in the introduction section, please add the objective of the study.

Author Response

Dear reviewer,

Thank you very much for your comments and professional advice. We have made the following modifications to the manuscript according to your suggestions:

Before the hypotheses in the introduction section, please add the objective of the study.

Response: Thanks for this kind recommendation. We added the objective of this study on page 2, lines 60-61.
